# Molecularly Targeted Lanthanide Nanoparticles for Cancer Theranostic Applications

**DOI:** 10.3390/nano14030296

**Published:** 2024-01-31

**Authors:** Guillermina Ferro-Flores, Alejandra Ancira-Cortez, Blanca Ocampo-García, Laura Meléndez-Alafort

**Affiliations:** 1Department of Radioactive Materials, Instituto Nacional de Investigaciones Nucleares, Ocoyoacac 52750, Mexico; guillermina.ferro@inin.gob.mx (G.F.-F.); alejandra.a.servicios@inin.gob.mx (A.A.-C.); blanca.ocampo@inin.gob.mx (B.O.-G.); 2Immunology and Molecular Oncology Unit, Veneto Institute of Oncology IOV-IRCCS, Via Gattamelata 64, 35138 Padova, Italy

**Keywords:** lanthanides, nanoparticles, targeting nanoparticles, nanoparticle toxicity

## Abstract

Injectable colloidal solutions of lanthanide oxides (nanoparticles between 10 and 100 nm in size) have demonstrated high biocompatibility and no toxicity when the nanoparticulate units are functionalized with specific biomolecules that molecularly target various proteins in the tumor microenvironment. Among the proteins successfully targeted by functionalized lanthanide nanoparticles are folic receptors, fibroblast activation protein (FAP), gastrin-releasing peptide receptor (GRP-R), prostate-specific membrane antigen (PSMA), and integrins associated with tumor neovasculature. Lutetium, samarium, europium, holmium, and terbium, either as lanthanide oxide nanoparticles or as nanoparticles doped with lanthanide ions, have demonstrated their theranostic potential through their ability to generate molecular images by magnetic resonance, nuclear, optical, or computed tomography imaging. Likewise, photodynamic therapy, targeted radiotherapy (neutron-activated nanoparticles), drug delivery guidance, and image-guided tumor therapy are some examples of their potential therapeutic applications. This review provides an overview of cancer theranostics based on lanthanide nanoparticles coated with specific peptides, ligands, and proteins targeting the tumor microenvironment.

## 1. Introduction

The group of lanthanides has been of interest for decades because of their luminescent properties [1,2]. In contrast to the other metals used to prepare nanoparticles, lanthanides are mostly found as dopant ions (Ln^3+^) in core–shell nanostructures and as pure lanthanide oxides [2,3,4,5,6]. Lanthanide nanoparticles (LnNPs) are suitable to be used as pharmaceutical forms for theranostic applications in cancer. For this purpose, LnNPs must be colloidally stable in biological fluids, exhibit negligible uptake in healthy cells and tissues, and have functional groups on their surface attached to antibodies, peptides, biomolecules, or ligands that provide specific molecular targeting to proteins present in the tumor microenvironment (TME) [4,5,6].

In the pharmaceutical industry, a colloidal system is defined as a system consisting of two or more phases, one of which is a fluid (liquid), and the other is dispersed in the form of fine solid particles. The constant Brownian motion of each dispersed particle surrounding the solvent shell is responsible for the stability of such colloidal solutions. Nanocolloids are high-molecular-weight particles (size 10–100 nm) that have difficulty crossing capillary membranes in healthy tissues (with an opening between endothelial cells of 2 nm), making them ideal for accumulation in tumors, where the intercellular space of the vascular endothelium opens up to 400 nm [7].

The size and surface properties of LnNPs can be easily modified to obtain stable colloidal solutions that meet pharmaceutical requirements. The multifunctionalization of LnNPs is a relevant advantage when used as molecular targeting probes, since TME is formed by a complex network of proteins/peptides/small molecules and immune cells, fibroblasts and cancer cells, all of which are potential diagnostic/therapeutic targets. The functionalization of nanoparticles confers high molecular affinity by creating multiple ligand binding sites on the nanosurface, which is necessary for targeting specific proteins expressed in TME.

The surface of LnNPs determines their interaction with body systems when administered in vivo, so when coated with biomolecules, they are not recognized as foreign by the immune system and are not toxic. Recently, it has been demonstrated that lutetium oxide nanoparticles coated with inhibitory peptides of fibroblast activation protein (iFAP) and prostate-specific membrane antigen (iPSMA) accumulate exclusively in the TME. Additionally, they are taken up by Kupffer cells and macrophages of the endothelial reticulum system without being trapped in the healthy parenchyma of the liver, lung, spleen or bone marrow. In other words, the elimination of functionalized Lu_2_O_3_-NPs from the body via the lymphatic system does not affect healthy tissues [5,6]. Furthermore, LaNPs can be prepared using pharmaceutical processes that comply with good manufacturing practices, which provides the added value required for clinical translation in theranostic applications [6].

Lanthanide-based nanomaterials offer multimodal approaches to diagnostics and therapeutics due to their magnetic, relaxivity, optical, and nuclear emission properties. The multifunctionality of LnNPs lies in their ability to provide diagnostic imaging of primary and metastatic tumors and their utility in photodynamic therapy, targeted radiotherapy (neutron-activated nanoparticles), drug delivery, and image-guided tumor surgery [8,9,10,11,12].

This review provides an overview of cancer theranostics based on lanthanide nanoparticles coated with specific peptides, ligands, and proteins for targeting the tumor microenvironment.

## 2. Physical Properties of Lanthanide-Based Nanoparticles for Theranostics

Metal nanoparticles (1–100 nm) have an extensive range of applications in biomedicine due to their physicochemical properties. They are utilized as targeted drug delivery, biosensors, as well as diagnostic and therapeutic agents for diseases like cancer. Therefore, an essential aspect is the synthesis route to achieve both the desired size and morphology. Additionally, these features are the starting point for successful surface modification and functionalization. The most effective methods for synthesizing these compounds include coprecipitation, solvothermal synthesis, thermal decomposition, microemulsion techniques, and wet-based chemistry methods [4,13].

The rare earth elements, also known as the lanthanides, are members of group IIIB of the periodic table, from Ln to Lu. Their electronic configuration can be denoted as [Xe]4f^(1−14)^5d^(0−1)^6s^2^, which implies that the 4f orbital is partially filled, except for lanthanum ([Xe]4f^0^) and lutetium ([Xe]4f^14^). The 4f electrons are responsible for the special properties of rare earths, including enhanced electronic, magnetic, and optical properties at the nanometer scale. Ln^3+^ is the most stable and frequently observed oxidation state for these elements [4,13,14,15].

Optically, these properties are favored by the large quantum numbers (*n* = 4, l = 3), which create partially allowed intraconfigurations at the 4f level and are further shielded from environmental effects by the screening effect of the electrons at higher energy levels (5s and 5p). Therefore, these materials offer numerous advantages, such as numerous absorption bands with narrow emission profiles from the UV-Vis to the NIR, excellent photostability, large Stokes and anti-Stokes shifts, long luminescence half-lives (µm to ms), luminescence quantum efficiency, low background autofluorescence (practically zero), no photobleaching line-shaped emission, and low biotoxicity [2,15,16,17,18,19,20].

A theranostic molecule possesses unique and versatile properties that make it suitable for both therapy and diagnosis. Lanthanide-based nanoparticles (Ln-NPs) exhibit these characteristics. Depending on the nanoplatform design, one of the following scenarios may occur [5,6,21,22,23,24].

Activation using near-infrared (NIR) light generates luminescence imaging for diagnostic purposes and triggers drug release for therapy.NIR activation also produces real-time luminescence imaging to evaluate the effectiveness of previously applied treatments for diagnosis and generates photothermal therapy (PTT) or photodynamic therapy (PDT).Neutron activation produces radioluminescence imaging with possible radiotherapy applications when beta particles are emitted.

Lanthanide nanoparticles include nanoparticles doped with lanthanide ions, alloys, metallic nanoparticles with varying percentages of lanthanides, lanthanide oxide nanoparticles, or core–shell nanoparticles [5,6,21,22,23,24].

### 2.1. Luminescence of Lanthanide-Doped Nanoparticles

Luminescence is a physical phenomenon generated by the movement of electrons contained in matter at different energy levels. It can be summarized as the absorption of energy to achieve an excited state which, upon returning to its basal state, releases this energy in the form of light, where the wavelength of the light emitted is a characteristic of the luminescent material and not of the incident radiation [17,21,25].

The luminescence of nanoparticles based on lanthanides occurs in the visible and NIR regions of the spectrum following stimulation by UV or NIR light. This phenomenon is further affected by both phonon energy and the strength of the crystal field that contains the Ln^3+^ ions. The use of Ln-NPs generates an optical image that enables visualization of subcellular morphological details and elucidation of signaling pathways and biological processes at the cellular level. Among the lanthanide ions, Eu^3+^, Dy^3+^, and Tb^3+^ exhibit the most effective luminescent characteristics [16,17,21,25].

### 2.2. Luminescence Emission Mechanisms

#### 2.2.1. Downshifting

Downshift emission is a nonlinear Stokes-shift process. It is based on the absorption of a high energy, short wavelength photon (NIR-I photon), which is emitted as a lower energy, longer wavelength photon (NIR-II photon), producing luminescence. The mechanism is based on the direct excitation of photons to an E2 state and then to an E1 relaxation state with the emission of nonradiative energy. Finally, it reaches the E0 state, its initial state, with the emission of luminescence. The nonradiative relaxation (E2→E1) is a multiphonon-assisted process governed by phonon dynamics. The generated emissions fall in the optical window between 700–1100 nm. These emissions can be of two types:Conversion of UV into visible light: The lanthanide ions representative of this emission are Er^3+^ (red emitter) and Tb^3+^ (green emitter).Conversion from UV-Vis to NIR: The lanthanide ions representative of this emission are Yb^3+^, Nd^3+^ and Dy^3+^.

In this process, as in the upconversion process, the structure and composition of the material, such as size, distribution, shape, and crystal phase, have a direct relationship to the quality and quantity of emitted luminescence [26,27,28,29,30].

#### 2.2.2. Upconversion

The upconversion process was first termed the infrared quantum counter in 1959. It is an anti-Stokes emission and a non-linear optical phenomenon. This process involves the absorption of two or more low-energy photons to produce the emission of higher-energy photons (shorter wavelength) to the incident photon via a high-duration energetic state [2,16,18,31].

Upconversion emissions result from four distinct energy transfer pathways: excited-state absorption (ESA), energy transfer upconversion (ETU), cooperative energy transfer (CET), and energy migration-mediated upconversion (EMU) [2,16,31,32,33].

ESA: This process involves sequential absorption of two or more low-energy photons by a single type of Ln^3+^ ion with medium-length energy states.ETU: In this process, there are two different luminescent centers, a sensitizer, and an activator. After excitation with a photon pump, energy is transferred from the sensitizer to the activator.CET: The photons generated have energies almost twice the transition energy. The emission energy originates from a significant disparity between the basal and the first excited state of the Ln^3+^ ion.EMU: In the core–shell structures, this procedure implicates four luminescent centers, including the sensitizer, activator, accumulator, and migrator. Energy is transferred consecutively across the interface of the core–shell.

The efficiency and process of upconversion exhibit high variation due to the varying energy levels in the 4f-4f intra-configurations. The complexity of these energy differences arises from the potential orbital-spin couplings of electrons and their interaction with the crystal field, resulting in process variations. Another influencing factor is the macrometric size (bulk material), where the absorption cross-section of the Ln^3+^ ions is small, which also generates a limited emission upconversion efficiency [25,31].

Lanthanide-based upconversion nanoparticles (Ln-based UCNPs) have improved the emission process, taking into account critical factors: (a) The symmetry of the Ln^3+^ ion in the crystal structure has been exploited to facilitate intraconformational transitions, with ions of smaller ionic radius than the lanthanoids strategically placed in the crystal; (b) Passivation of the surface of core–shell structures in order to minimize the quenching effect due to surface defects; (c) Modulation of transfer energy to reduce non-radiative losses and enhance radiative emission is ideal. This can be achieved through the use of metals that have a d–d transition in their crystal structure [2,25].

Ln-based UCNPs effectively transform various photons from the near-infrared (NIR) region, with low energy, into high-energy NIR, visible, or ultraviolet photon emissions [19,21]. Excitation with near-infrared (NIR) light (980–808 nm) within the biological optical window (700–1000 nm) enables deeper penetration into biological tissues and subsequent imaging with high sensitivity. Excitation with this light produces minimal autofluorescence because the NIR light does not excite fluorophores in the organism, resulting in a near-zero signal-to-noise ratio [2,5,17,18,19,20,21,34]. These systems serve as therapeutics through three approaches:Photodynamic therapy (PDT): a non-invasive therapy for cancer treatment with three essential components—light, photosensitizer, and oxygen. PDT involves the NIR light irradiation of UCNPs to generate upconversion emission, which excites the photosensitizer (PS). Subsequently, the energy from the excited PS is transferred to nearby triplet oxygens (3O_2_), resulting in the creation of singlet-type reactive oxygen species (ROS) responsible for damaging cancer cells (O_3_). This therapy yields better effects at shorter distances between the activator donor (energy donor) and the PS (energy acceptor); it is low-cost, accurate, and has minimal long-term side effects [5,16,19,21].Photothermal therapy (PTT): Therapy that converts light into heat to generate local hyperthermia to cause cancer cell death. The therapy is typically generated using AuNPs, organic dyes, graphene oxides, or QDs [5]. Its mechanism is based on multiphoton relaxation of the excited states of trivalent Ln^3+^ ions, combined with emission quenching effects generated by nonradiative centers located in the periphery [5,21].Drug delivery and therapy: Ln-based UCNPs enable drug delivery and release from drug-carrying platforms by functioning as high-penetration probes without interfering with the therapeutic process of the drugs. Additionally, photoactivation or photorelease at specific sites following noninvasive stimulation of the UCNPs with light, triggers drug release. The major advantage of the therapy is the use of NIR light, which avoids unwanted phototoxic tissue damage, in contrast to the use of UV light [16].

On the other hand, Ln-based UCNPs can function as multimodal platforms with the right design, as they are easily adjustable (Table 1).

#### 2.2.3. Quantum Cutting

This phenomenon is based on the absorption of high-energy photons emitted as two or more lower-energy photons. The efficiency of this phenomenon is greater than 100%. An example of this emission is presented by co-doped lanthanide compounds Gd^3+^ or Yb^3+^/Tb^3+^, Pr^3+^, Ho^3+^ or Dy^3+^.

Gd^3+^ or Yb^3+^ are the acceptor ions that emit luminescence by transferring energy to one of the complementary ions by multiphoton relaxation in the NIR. An option for this structure is hybrid NPs (organic–inorganic) with a sensitizer to absorb UV light. It can be said that this process is a mixture of downconversion and upconversion mechanisms. However, the clear difference between the first two and QC is that the latter has multiphoton emission [28,43].

A schematic description of the spectral conversion mechanism of the different designs of Ln-based nanoparticles is shown in Figure 1.

### 2.3. Photoluminescence

The primary restriction of Ln-based NPs for generating luminescence is the low absorption coefficient of lanthanide ions. While the proximity and concentration of the Ln^3+^ ions are enhanced by nanometer size, incorporating sensitizing molecules is the best approach to increase the absorption coefficient. These molecules usually form stable complexes by binding to the Ln^3+^ ions. They efficiently absorb excitation energy, which is then transferred to the lanthanide ion when it returns to its basal state. The lanthanide ion uses this transferred energy to generate luminescence [44,45].

If the sensitizing molecules are organic, their absorption efficiency is due to the presence of π bonds, and the energy is absorbed through π–π* transitions. Conversely, if they are inorganic molecules (e.g., phosphonates), their absorption efficiency is mainly due to dipole–dipole, dipole–magnetic, or dipole–electric transitions [44,46].

The most representative instance of this phenomenon is provided by UCNPs, which emit photoluminescence (PL) in both the visible and infrared (IR) regions upon being excited by IR light. The properties are favored by characteristics such as size, surface components, crystallinity, chemical composition, involved fluorophores, and the heterogeneity of electronic states among different components [27,47].

### 2.4. Radioluminescence

X-rays are high-energy electromagnetic radiation with short wavelengths and high penetrating power. They are currently used for X-ray imaging as a low-cost, high-availability diagnostic tool. However, one of its major drawbacks is that it provides a two-dimensional image, which results in loss or lack of relevant information. In other words, the technique has low sensitivity [48,49].

On the other hand, when ionizing radiation interacts with specific materials, it can produce radioluminescent optical photons. Materials with a high atomic number have demonstrated the greatest efficacy due to their ability to absorb X-rays. This interaction can generate Cerenkov radiation, interact with scintillators, and induce fluorescence, phosphorescence, and luminescence. Nanoparticles have several biomedical applications, including phototherapy, drug delivery, radiotherapy monitoring imaging, and molecular imaging [50,51].

Nanoparticles, particularly those based on lanthanides, have gained attention as materials with potential for radioluminescence imaging. They are influenced by the presence of a significant number of atoms on their surface, as well as by the quantum confinement of electronic states and the area-to-surface ratio. The aforementioned results in an increase in the interaction rate between ionizing radiation and ions, leading to more efficient radioluminescence generation despite their small size. Chemically, the high-Z, cross section, and high crystallinity of the ions are critical factors that enhance X-ray absorption and subsequent conversion into UV or NIR light. Last but not least, the radiation dose received or available to the nanoparticles in their vicinity is enhanced by the phenomenon known as radiosensitization [5,50,52]. However, the small size of nanoparticles when compared to ionizing particles has been deemed a challenge for achieving the desired interaction, resulting in the excitation of only a specific fraction of nanoparticles. Nevertheless, when the same nanoparticles (radionanoparticles) are the emitters of ionizing radiation, this gap is enhanced, as is the intensity of the luminescence emitted by the nanomaterials. The excitation is also “preserved” and influenced by the half-life of the emitting radionuclide [22,33,50,53].

Another benefit of Ln-based nanoparticles is the presence of their surface functionalization components, such as biomolecules, dyes, targeting ligands, and small peptides. These components typically have highly energetic absorption bands, allowing them to contribute to luminescence through energy transfer between their d electrons and the 4f electrons of the Ln^3+^ ions [22,23,50,53].

Regarding the therapeutic use of radioluminescence, some reports describe the combination of scintillator materials with radiotherapy to promote photodynamic therapy [53,54,55,56]. The approach proposes converting X-rays to light via the use of a scintillator to activate the photosensitizer and achieve a therapeutic effect. The phenomenon’s mechanism is described as relying on the transfer of Cerenkov radiation energy by a photon–photon or beta–photon emission interaction. The primary benefit of this approach is the restriction of additional dose radiation. However, further advancements and research are needed to address the current limitation in the energy transfer of the scintillators [50,57].

### 2.5. Fluorescence Imaging in the Second Near-Infrared Biological Window (NIR II 1000–1700 nm)

Optical stimulation in the near-infrared (NIR, 700–1700 nm) window has advantages in biomedical imaging because of its low absorption, low scattering, and minimal tissue autofluorescence, which is most evident in the NIR-II region (1000–1700 nm) [58,59].

NIR-II region is very close to zero at wavelengths longer than 1300 nm [60,61,62]. Chemically, the emitting and receiving electrons must conserve their spin moment to enable fluorescence emission. Physically, fluorescence is governed by the anti-Stokes shift phenomenon. NIR light excites the materials and produces emission in the NIR II region. In addition, by the same process, NIR light also excites sources for visible light-emitting fluorophores [24,54,55,63].

Some of the applications of fluorescence imaging with nanoparticles include gene detection, protein analysis, evaluation of enzyme activity, tracking of elements, diagnosis of early-stage diseases, and monitoring of therapeutic effects in real-time. The success of these applications is heavily reliant on several factors: (a) Employ excitation and emission wavelengths within the NIR region, preferably NIR-II, to enable deeper penetration of optical photons with less excitation damage to other components; (b) Implement systems with high biocompatibility and adequate functionalization and stability; (c) Ensure that fluorescence intensity and duration are maintained; (d) Avoid rapid scavenging when using nanoparticles. The appropriate size ranges from 5.5 to 150 nm, depending on the pathology to be analyzed [61,62]. Among the commonly used nanomaterials are fluorescent nanocrystals, or QDs, which emit light from 380 to 2000 nm. These QDs include Ag_2_S, Ag_2_Se, ZnS, PbS, PbSe, and CdHgTe. Single-walled carbon nanotubes (SWCNTs) and Ln-based NPs are also frequently utilized.

Ln-based nanoparticles (NPs) can emit in NIR II (Stokes-shift-based imaging agents). This phenomenon is due to the optical and electronic properties of electrons in the 4f orbital. Host matrixes are the primary components, along with sensitizers and activators, where the latter two contain Ln^3+^ ions. These systems provide benefits such as deep tissue penetration, reduced background noise, and lower toxicity. Other features include high temporal resolution (20 ms), high spatial resolution, and high tissue penetration (up to 3 cm) [28,54,56,60,63]. Moreover, the luminescence efficiency in Ln-based nanoparticles is determined by both internal and external factors. The internal factors are related to the Ln^3+^ ion, including the resonance level and intrinsic efficiency of the electric–dipole transitions of its 4f electrons. External factors include other competitive upconversion processes and non-radiative cross-relaxation, among others [28].

Examples of Ln-bases NPs include Yb^3+^-based nanoparticles with NIR-II luminescence at 1525 nm, Nd^3+^-based nanoparticles with NIR-II luminescence between 1300–1400 nm, Er^3+^-based nanoparticles with NIR-II luminescence between 1500–1700 nm, and core@shell type organic dye-based nanoparticles with Ln^3+^ dopant ions [28,59,64,65].

### 2.6. Magnetic Resonance Imaging

MRI is based on the net polarization of the nuclear spins of water protons in the presence of an intense magnetic field of 1.5 to 3T. Therefore, the magnetic properties of the contrast agent require the insertion of one or more metal centers with unpaired electrons [66]. Gadolinium, terbium, dysprosium, and holmium oxide nanoparticles are of particular interest because they have appreciable magnetic moments, which is useful for MRI [67]. Gd^3+^ ions have been widely used to produce molecular T1 contrast enhancement [66]. Many papers have been published on Gd^3+^-based nanostructures of different composition, shape, and size. However, only a few of them have been functionalized with small molecules, such as folic acid [50,54], RGD, chlorotoxin and transactivator of transcription (TAT) peptides [68,69,70], and the anti-thrombomodulin antibody [71]. Other Ln^3+^ ions increase the magnetic moment as the magnetic field strength increases before reaching saturation, resulting in higher transverse relaxivities [72].

Tb_2_O_3_, Ho_2_O_3_, and Dy_2_O_3_ nanoparticles have been reported as useful as T2 MRI contrast agents at high MR fields in vivo [71,72]. Although the lanthanide ion Eu^2+^ also has seven unpaired electrons in its outer electron shell, it has a larger ionic radius than Gd^3+^, which results in a faster water exchange rate. This characteristic ensures that Eu-based contrast agents have relatively high relaxivity values [66]. However, to date, no functionalized nanoparticles containing Tb^+3^, Ho^+3^, Dy^+3^, and Eu^2+^ ions have been described for molecularly targeted MRI.

## 3. Modification of Lanthanide-Based Nanoparticles with Active Molecules for Theranostics

Lanthanide nanoparticles have been functionalized with various biomolecules to molecularly target different specific sites in the tumor microenvironment (Table 2 and Figure 2).

### 3.1. Folic Acid (Small Molecules)

The first attempt to produce targeted NPs with biological molecules was made using stable and non-immunogenic small molecules such as folate or folic acid (FA), a water-soluble vitamin with a low molecular weight (411.4 Da). Folate receptors are overexpressed on rapidly proliferating cells, such as cancer cells, due to their increased need for folate compared to normal cells. This overexpression in human tumors such as endometrial, breast, ovarian, and brain cancers can be used to target these tumors [65].

Cao et al. were the first to report water-soluble UCNPs functionalized with FA by introducing 6-aminohexanoic acid (AA) and oleate to control generation and crystal growth. AA was attached to the lanthanide ions through the carboxylic acid group, providing free amine groups that made the NPs dispersible in water and allowed conjugation with the FA. The UCNPs showed intense luminescence and helped obtain mouse lymphatic capillary imaging with a high signal-to-noise ratio. Their FA-functionalized UCNPs showed specific internalization in KB (FR-positive) cells, but the author did not test them in vivo [73]. The first FA-functionalized UCNPs used in vivo were reported by Idris et al., who used them as nanotransducers to convert NIR light to visible wavelengths and also as PS. The authors showed that irradiation of these UCNPs with a 980 nm laser produces green and red light and activates the coloaded PS cargo, increasing the generation of cytotoxic singlet oxygen and reducing cell viability in vitro, and also demonstrated growth inhibition of melanoma tumors in mouse models [74].

Sun et al. prepared three kinds of FA-functionalized UCNPs (UCNC-FA) by doping NaYbF_4_ with Er^3+^ and Tm^3+^ ions (UCNC-Tm-FA, UCNC-Er-FA and UCNC-Er, Tm-FA) and demonstrated that UCNC-FA NPs were able to efficiently target HELA cells in vitro with low toxicity and good biocompatibility, and thus could be used as a potential fluorescent contrast agent [75]. Similar results were reported by Gainer et al. using NaYbF_4_ doped with Er^3+^ and Yb^3+^ and functionalized with FA; they were able to obtain luminescence images of ovarian cancer by multiphoton scanning microscopy, demonstrating the potential application of lanthanide nanoparticles to detect relevant receptors in tissues [76]. NaYbF_4_ doped with Yb^3+^, Tm^3+^ were also used by An et al., but they incorporated the Fe^3+^, and demonstrated that the addition of 20% of the non-luminescent Fe^3+^ increased the NIR UCL intensity 20-fold. The authors also add NaGdF_4_ to the nanoparticle to develop multimodal in vivo bioimaging, as Gd^3+^ ions exhibit high X-ray attenuation and strong paramagnetic properties. They demonstrate that FA functionalization of this NP can successfully target tumors in vivo and obtain trimodal imaging (fluorescence, MRI, and X-ray) [77]. Recently, NaYbF_4_ core shells doped with Nd^3+^ and functionalized with FA have also been shown to be promising dual-PS nanoconstructs for the treatment of deep-seated tumors, as they can maximize the antitumor efficacy of PDT through the 808 nm laser excitation capability [78]. Zeng et al. also enhanced the therapeutic efficacy in deep tumor sites producing ROS and PTT by irradiation of 980 nm laser irradiation of a Linde Type A (LTA) zeolite-derived UCNPs dopped with ER^3+^ and Yb^3+^ functionalized with FA-PEG [79].

Metal-organic frameworks (MOFs) are also promising candidates for drug delivery due to their microporous structure. UCNPs have been used to increase drug concentration at the tumor site by encapsulating molecules and rendering them biologically inert, thereby increasing chemotherapeutic efficacy and reducing toxicity and side effects for patients. By physical adsorption, Liu et al. synthesized PeGylated UCNPs loaded with the anticancer drug doxorubicin (DOX). They conjugated them with folic acid to evaluate the loading and release of DOX at different pH values. They found that 20% of DOX was released at physiological pH (7.4), with this rate increasing at lower pH values. In vitro studies showed that this nanoconstruct produced cytotoxic effects in folate receptor-positive cells due to the enhanced cellular uptake of DOX caused by the specific FA targeting [80]. Chien et al. reported another folate-PEGylated UCNP as a delivery agent for bioimaging and chemotherapy. Still, in this case, DOX was thiolated on the surface of these NPs by disulfide bonds that can be cleaved by the lysosomal enzymes present in the cell [81]. When UCNPs were irradiated with a 980 nm laser, they emitted upconversion luminescence at 360 nm, which uncaged the photolabile o-nitrobenzyl group and allowed FA to target the tumor cells and deliver the DOX via FR-mediated endocytosis. This nanostructure’s therapeutic efficacy and in vivo, photoluminescence imaging was demonstrated in mice bearing a HeLa tumor model. The brightest NIR emission in the tumor was found 5 h after injection, and mice showed significant inhibition of tumor growth compared to control groups. Recently, Cong et al. used a new layer-by-layer self-assembly method to synthesize core–shell microspheres NaYF_4_:Yb,Er@NaYF_4_:Nd@MIL-53, and encapsulated the FA into the structure to be used as DOX delivery carriers [82]. The in vitro test showed that the core–shell was non-toxic to cells, and the functionalized version could recognize the target cells.

Cao et al. synthesized a MoS_2_ quantum dots (QDs) decorated UCNPs nanoplatform, modified it with PEG and FA, loaded it with DOX (UMPFD), and evaluated it as a tumor-targeting agent for chemo-photodynamic synergistic therapy [83]. These constructed UCNPs were designed to convert 980 nm NIR into visible light, which can activate the MoS_2_QDs to produce oxygen species by fluorescence resonance energy transfer. UMPFDs exhibited good biocompatibility and tumor targeting with pH-responsive drug release and abundant ROS generation.

Mesoporous silica nanospheres (MSNs) have also been functionalized with FA for tumor targeting. Shi et al. reported MSNs loaded with Zn1.1Ga1.8Ge0.1O4:Cr^3+^, Eu^3+^ to obtain persistent luminescent nanoparticles (PLNPs), which were modified with FA and subsequently loaded with DOX [84]. The DOX-NLPLNPs@MSNs-FA was evaluated as a real-time monitoring drug delivery agent. The results showed that this construct exhibited long afterglow NIR luminescence and high sensitivity to target tumors in vitro and in vivo with a sustained release of DOX.

Lanthanum oxide nanoparticles have also been used as optical devices in medicine. Recently, hafnium dioxide (HfO_2_) NPs doped with Eu^3+^, Gd^3+^, and Tb^3+^ ions and functionalized with FA were synthesized by different routes to evaluate their potential as a matrix for multimodal theranostic agents for image-guided radiotherapy. The results showed that sol–gel-based synthesis was the best method to prepare uniformly doped particles with good size control. It was also demonstrated that introducing lanthanide ions could tune the luminescence, MR, and CT properties. Gd:HfO_2_ NPs showed the best properties, such as the lowest degradation rate and no relevant in vitro cytotoxicity [85]. Recently, Cai et al. reported a versatile nanosystem for real-time monitoring of cancer treatment efficacy. They developed gadolinium oxyfluoride (GdOF) nanoparticles doped with Yr and Er, which were used to encapsulate Au nanosheets, then decorated with both DOX and the photosensitizer rose bengal (RB), and functionalized with FA [54]. Thanks to doped lanthanide ions and encapsulated Au nanosheets, trimodal fluorescence, CT, and MRI imaging were obtained simultaneously. Moreover, the fluorescence resonance energy transfer from GdOF to RB in this nanosystem contributed to the high PDT efficacy, which acted synergistically with the chemotherapy in the targeted tumor microenvironment, resulting in tumor eradication after 14 days of administration in mice with cervical carcinoma.

### 3.2. Peptides

Peptides are cancer-specific ligands with moderate size, high stability, and low immunogenicity but still possess a high specificity towards their target. Moreover, they are easy to synthesize on a large scale and modify; therefore, some peptides have been used to functionalize the NPs to be applied as image-guided tracking and also to direct chemical drug-loaded lanthanide nanoprobes to the tumor sites, achieving higher tumor inhibition than pristine drugs [9].

The most commonly used peptide to functionalize NPs is arginine-glycine-aspartic acid (RGD) because it can recognize the adhesion receptor integrin α_v_β_3_, which is overexpressed on tumor endothelial cells [104]. Xiong et al. reported the first UCNPs doped with Yb, Er, and Tm and functionalized with RGD for fluorescence-targeted imaging in vivo. To improve the blood circulation of the nanosystem, the authors added a PEG linkage [91]. Confocal imaging of tissue sections at high penetration depths of these nanosystems showed that UCL imaging without an autofluorescence signal was possible. In addition, the experimental results in mice bearing human glioblastoma (U87MG) showed that the functionalization of this nanoplatform enables target-specific imaging of tumors with a high signal-to-noise ratio, making this technique also suitable for tracking in vivo components of biological systems. Jin et al. added Ga^3+^ DOTA to RGD-functionalized UCNP to perform in vivo fluorescence upconversion studies and MRI of glioblastoma with excellent results in mice bearing xenografted tumors [68]. Tang et al. used similar nanosystems to deliver the chlorin e6, a photosensitizer, and a chelator for Mn^2+^ loaded in human serum albumin, conjugated to the UCNP. The results demonstrated these NPs’ potential for MRI and glioma PDT [92]. Recently, a NIR-regulated nanoplataform called UCNP@TTD-cRGD was synthesized by using an amphiphilic polymer to encapsulate UCNPs and the TTD luminogen and functionalized with cyclic RGD (cRGD) to be used as an antitumor and bioimaging agent. The results showed that this nanosystem could selectively target MDA-MB-231 cancer cells in mice bearing subcutaneous tumors and inhibit their growth due to two main facts: the NIR-regulated PDT treatment in vivo and also the ROS generation by the close matching of UCNP emission and the absorption of the aggregation-induced emission [93]. CsLu_2_F_7_-based UCNPs doped with Yb, Er, and TM were also functionalized with RGD and loaded with the chemotherapeutic drug α-tocopheryl succinate (alpha-TOS) and the photothermal agent ICG for multifunctional imaging (UCL and CT) and targeted synergistic therapy. In vivo studies in mice bearing U87MG tumors showed a significant tumor reduction after treatment with this nanoplatform [94].

Prostate-specific membrane antigen (PSMA protein) is an important molecular target due to its overexpression in various cancer cells, including advanced and metastatic prostate cancer. This has led to successfully generating radiolabeled PSMA inhibitor peptide (iPSMA) based systems as agents for molecular imaging and radiotherapy [23,105,106]. Another protein overexpressed in several neoplastic cells, including early prostate cancer, is the gastrin-releasing peptide receptor (GRPr), which recognizes the bombesin peptide selectively and specifically [107]. Therefore, a heterodimeric peptide of iPSMA and bombesin has been reported in the functionalization of [^153^Sm]Sm_2_O_3_ nanoparticles with radioluminescent and radiotherapeutic features (NIR at 750 nm), allowing in vivo optical imaging of their biodistribution and high affinity with respect to PSMA protein and GRP receptors [22].

The iPSMA peptide has also been used to functionalize [^166^Dy]Dy_2_O_3_, ^166^Ho_2_O_3_, and [^177^Lu]Lu_2_O_3_ nanoparticles with utility as targeted radiotherapy systems [23,105,106]. Dysprosium and holmium oxide nanoparticles were reported as a novel in vivo generator. That is, [^166^Dy]Dy_2_O_3_ NPs were obtained by neutron irradiation of ^164^Dy_2_O_3_ NPs, which decay to ^166^Ho_2_O_3_ NPs under a nuclear transient equilibrium. After functionalization with the iPSMA peptide, an in vivo [^166^Dy]Dy_2_O_3_-iPSMA/^166^Ho_2_O_3_-iPSMA generator for medical applications was obtained. In vitro and in vivo studies demonstrated the potential of this molecularly targeted lanthanide generator to deliver ablative radiation doses to HepG2 liver cancer cells [89].

On the other hand, chemical analysis of Lu_2_O_3_-iPSMA also resulted in a well-defined hemispherical shape with a uniform and monodispersed size distribution (30–45 nm) and characteristic radioluminescence features after neutron activation. The in vitro studies showed a high affinity of ^177^Lu_2_O_3_-PSMA for PSMA-expressing cells. Preclinical biodistribution in murine models demonstrated the nanosystem’s potential for in vivo nuclear and optical imaging and its utility for targeted radiotherapy [22,23].

Fibroblast activation protein (FAP) has become one of the most important molecular targets involved in the tumor microenvironment [108]. Cancer-associated fibroblasts induce the cancer phenotype, account for 90% of the macroscopic tumor mass, and strongly express FAP in >90% of carcinomas. As a result, FAP is overexpressed in the neoplastic microenvironment of more than ninety percent of epithelial tumors and is a major contributor to cancer progression [108]. This is why Lu_2_O_3_ nanoparticles activated by neutron activation have also been functionalized with FAP inhibitor peptides (iFAP). Additionally, the iPSMA and iFAP peptides were concomitantly attached to the lutetium oxide nanoparticles to be used as a dual-targeted radiotherapy nanosystem [5,6]. The quality control results showed that the nanoparticles with the appropriate pharmaceutical properties were produced reproducibly under good manufacturing practices [6].

Bombesin can also be recognized by GRPr overexpressed in breast, prostate, lung, CNS, colon, and pancreatic tumors, where it functions as an autocrine growth factor [109]. Tang et al. first used bombesin to functionalize Gd^3+^-loaded UCNPs and used it to obtain in vivo MRI, CT, and UCL images of prostate tumors [88].

Another peptide that has been used to functionalize lanthanide nanoparticles is Angiopep-2 (Ang2), a peptide designed from the Kunitz domain that binds to low-density lipoprotein receptor-related protein 1 (LRP1) at the blood–brain barrier (BBB) and has been shown to enhance nanosystem accumulation in brain tumors [110]. UCNPs were coated with oleic acid and functionalized with PEG-conjugated Ang2 peptide to deliver hydrophobic photosensitizers to brain tumors in mouse models [87]. In vivo studies showed that ANG-IMNPs could cross the BBB and selectively deliver synergistic PDT and PTT to brain astrocytoma tumors, significantly improving mouse survival. Ang et al. developed a multi-shell Nd-UCNPs, loaded with DOX and chlorin e6 (Ce6) and functionalized with an Ang2 peptide for a synergistic metronomic chemo-photodynamic therapy of glioblastoma [86]. This study provided evidence that these nanosystems could cross the BBB and be endocytosed by the lysosomes of glioblastomas in mouse models. In addition, they demonstrated a 3.5-fold higher antitumor efficacy compared to standard chemotherapy and inhibited angiogenesis in the tumor microenvironment. Ren et al. used down-conversion nanoparticles (DCNPs) modified with a dye-brush polymer and functionalized with Ang2 peptide. They delivered them into the glioma by temporally opening the BBB using ultrasound sonication [64]. The authors obtained a targeted NIR IIb fluorescence imaging with a very high tumor-to-background ratio (TBR = 12.5) for small orthotopic gliomas. They found that the glioma size obtained by this method was very close to the size obtained from T2-weighted MRI images.

KE108, a somatostatin analog that binds with high affinity to all five somatostatin receptors, has also been used to specifically deliver nanoplatforms to neuroendocrine tumors [111]. Chem et al. developed theranostic micelles formed by UCNPs functionalized with photosensitive copolymer (PNBMA-PEG) and the photosensitizer RB. The micelles were loaded with the anticancer drug AB3 and conjugated with the KE180 peptide. These UCNP micelles proved to be good UCL imaging agents without apparent systemic toxicity. Moreover, they offer the possibility of selectively targeting neuroendocrine tumors in vivo, producing synergistic effects of chemotherapy and PDT, resulting in higher antitumor efficacy than a single technique [90].

TP53 is one of the most frequently mutated genes in human cancer; therefore, small molecules capable of reacting with the mutant p53 function have recently been used as an anticancer strategy [112]. The p53-activating peptide PMI is one of them. He et al. designed a nanorod using lanthanide-doped nanocrystals functionalized with PEG and PMI to treat human colon cancer in mouse models. This LProd proved to be useful for real-time disease tracking and produced tumor growth inhibition by activating the tumor suppressor p53, showing improved tumor targeting compared to its spherical counterpart [113].

It has been reported that the dual-targeted radiotherapy strategy increases the affinity of the nanosystem for the tumor sites. Proof of this fact is the lanthanide oxide nanoparticles functionalized with heterodimeric peptides such as bombesin/iPSMA or iFAP/iPSMA described above [5,6,23]. Recently, a dual-targeting nanoplatform (UCNP@P-RGD-NGR) has also been developed using polydopamine-coated UCNPs conjugated with two peptides, the RGD and the asparagine-glycine-arginine (NGR), which were used to target the integrin and aminopeptidase N receptors present in tumor cells. This nanoplatform was shown to be non-toxic, biocompatible, and highly specific for lung tumor cells in vitro and in vivo, as well as be able to discriminate tumors from the surrounding normal tissue by UCL [95]. Zha et al. also used this dual targeting to monitor and inhibit cancer associated with the Epstein–Barr virus using UCNPs functionalized with two peptides that specifically bind to EBNA1 and LMP1, two overexpressed EBV oncoproteins. This nanosystem was used as a pH-sensitive imaging agent for cancer differentiation, showing enhanced specific uptake in EBV-positive tumors, resulting in high therapeutic efficacy demonstrated by in vivo tumor inhibition [96].

### 3.3. Antibodies

Several antibodies or antigens have been attached to lanthanide nanoparticles to produce luminescent images and/or targeted radiotherapy. In these approaches, the role of antibodies is primarily to target approaches to the interested tissue through specific recognition. In these terms, antibody-functionalized lanthanide-doped CaF_2_ biolabel for cancer cell targeting was synthesized by Sasidharan et al. They prepared monodispersed lanthanide (Eu^3+^) doped CaF_2_ nanoparticles and functionalized them with anti-EGFR through EDC-NHS coupling chemistry for specific binding to EGFR-overexpressing cells. The nanoparticles exhibited strong fluorescence emission at 612 nm [97]. EGFR overexpression has also been used to enable specific binding of functionalized UNPs (anti-EGFR-UNPs) for imaging modalities to achieve three-dimensional functional tissue imaging based on laser scanning in cancer cells [98].

Lanthanide nanoparticles functionalized with polyethyleneimine (PEI) and an antibody to detect alpha-fetoprotein (AFP) were also prepared. The positively charged PEI-PLNPs were bound to the negatively charged antibody-functionalized gold nanoparticles, forming an FRET inhibition probe. The persistent luminescence of the PEI-PLNPs was quenched in the presence of AFP but recovered when the AFP-specific antibodies were desorbed from the nanoparticles [99]. In a different nanoparticle–antibody system for targeted radiotherapy, the mAb-201b was conjugated to Au-coated lanthanide phosphate nanoparticles containing ^177^Lu to target thrombomodulin receptors for pulmonary metastatic disease. This interesting approach resulted in up to 95% of the injected dose being delivered to the lungs [71].

Fluorescent lanthanide oxyfluoride nanoparticles conjugated with anti-CD-33 antibody for potential treatment of acute myeloid leukemia by conjugating PMI, a dodecameric peptide antagonist of MDM2 and MDMX and a CD-33-targeted humanized monoclonal antibody to nanoparticles via metal thiolate bonds. NP emitted fluorescence at 545 nm with minor red emission peaks at 585 and 620 nm. The lanthanide nanoparticles used in this study were composed of LaOF, Ce, and Tb and were designed to have specific properties for their application in drug delivery for acute myeloid leukemia (AML) therapy through binding to CD33 surface receptor overexpressed in AML, which allows the intracellular drug delivery of PMI. This peptide antagonizes MDM2 and MDMX to activate the p53 pathway, leading to apoptosis of AML cells. This nanosystem also enables real-time visualization of apoptotic events in AML cells [100].

Biocompatible cubic-phase erbium-based rare-earth nanoparticles (ErNPs) with downconversion luminescence at 1600 nm for dynamic imaging of cancer immunotherapy were also prepared. Functionalization was performed with cross-linked hydrophilic polymer layers decorated with anti-PD-L1 antibodies for molecular imaging of PD-L1 in a colon cancer model, achieving a tumor-to-normal tissue ratio of approximately 40. The PD-L1 antibody enabled the specific targeting and imaging of PD-L1 in the tumor microenvironment to evaluate the immune checkpoint blockade therapy and the response to immunotherapy [101].

NP-avidin and NP-IgG were synthesized by Eliseeva et. al. The lanthanide (Eu, Gd) binuclear helicate [Ln_2_(LC_2_)_3_] was embedded into silica nanoparticles Ln_2_(LC_2_)_3_]@SiO_2_/NH_2_ used to detect and mark cancerous cells in immunocytochemical assays. Ln_2_(LC_2_)_3_]@SiO_2_ was then conjugated with either avidin or goat anti-mouse IgG antibody to target mucin-like antigens expressed specifically on the cell membrane. The nanoparticles display red emission and have a high quantum yield, making them effective markers for cancer cells. The bioconjugates of the nanoparticles with the IgG antibody show a higher luminescent signal-to-noise ratio compared to the bioconjugates of the molecular probe with IgG [114].

Biosensors traditionally focus on the use of antibodies for antigen detection. However, antibody detection is now increasingly utilized in biosensor research. Antibody detection has been applied to monitor immune responses to biotoxins based on the antigen–antibody recognition systems in HIV infections, tuberculosis, human papillomavirus, and breast cancer or to measure the vaccine-induced immune response. Covalent conjugation of okadaic acid to LaF_3_:Ce, Tb nanoparticles was performed to detect anti-okadaic acid antibodies. This system allows luminescence resonance energy transfer (LRET) to detect anti-okadaic acid rabbit polyclonal IgG for new antigen–antibody recognition applications [115].

### 3.4. DNA/RNA and Aptamers

Attaching DNA/RNA to nanomaterials will enable nucleic acid-based assembly and drug delivery. RNA interference (RNAi) is a natural cellular process of post-transcriptional gene regulation that uses small, double-stranded RNAs to induce the controlled silencing of genes (siRNA). Therefore, by introducing siRNAs, we are able to harness the RNAi machinery for the therapeutic control of genes and the treatment of a variety of diseases [116,117,118].

For siRNA delivery, nanoparticles have been recently proposed as excellent approaches. siRNA can be incorporated into an NP formulation by covalently binding to NP components or by electrostatic interaction with the NP surface [119]. Coordination of lanthanides with phosphate groups and nucleobases is used to construct lanthanide-doped nanoparticles. The siRNA/NaGdF_4_ nanoparticle system allowed siRNA to escape from the endosome for efficient gene silencing in vivo. In addition, siRNA targeting PD-L1 (siPD-L1) complexed with NaGdF_4_ was able to block PD-L1 expression and thereby suppress tumor growth in mouse models of breast and colon cancer [102].

Aptamers are single-stranded DNA or RNA sequences between 20 and 80 nucleotides in length that bind to their target with high affinity. The AS1411 aptamer (dsDNA) binds nucleolin overexpressed in the nucleus of cancer cells and was employed to synthesize UCNPs@PDL@dsDNA/DOX for doxorubicin delivery into the nucleus of cancer cells [103].

## 4. Theranostic Lanthanide Nanoparticles with Potential for Clinical Translation

The stability of the colloidal solution is the first requirement for the clinical translation of the different theranostic nanosystems of LnNPs as stable, effective, and safe drug forms [115].

For the stability of LnNPs, repulsive forces must be generated by similar charges to prevent the coagulation of the colloidal particles. After preparing a nanocolloidal solution from which all ions are removed by dialysis, the particles agglomerate because the total surface area is reduced, the particle size increases, and the particles settle rapidly. Therefore, the presence of electrolytes that provide some electrical charge is always necessary. However, the number of electrolytes must be limited to the amount that the colloids can adsorb, since an excess reduces the zeta potential by accumulating ions opposite to the charge of the LnNPs. Therefore, the number of electrolytes should be sufficient to keep the zeta potential below a critical level (Table 3) [120].

In this context, most of the lanthanide nanoparticles functionalized with small molecules, peptides, antibodies, and oligonucleotides mentioned in Table 2, have demonstrated that they can be prepared as isotonic solutions with high colloidal stability, potentially satisfying the first requirement for a stable dosage form.

The stability of nanoparticles as a pharmaceutical form with high zeta potential values is a prerequisite for their in vivo application. However, interactions with components of biological fluids, such as proteins, electrolytes, and small molecules, influence their electrical potential and reduce the absolute value of the zeta potential, although without affecting the stability of the nanosystems when coated with biomolecules covalently bound to their surface, which guarantees colloidal stability [23].

The development of efficient processes that meet the good manufacturing practices (GMP) requirements of regulatory authorities is one of the most important current challenges in the production of theranostic nanoparticles for use in patients. Biomolecule-coated LnNPs (Table 2) can be produced under GMP procedures as sterilized colloidal liquids with the quality characteristics of a clinical-grade formulation [6].

The toxicity associated with nanoparticles varies depending on their intrinsic composition, shape and size, surface area, surface layer, and in vivo stability [121]. The mechanism of nanoparticle toxicity is based on their ability to generate cytotoxic and genotoxic reactive species related to immune-mediated effects (inflammation) and oxidative stress [122]. Among the most studied are the toxic effects of metal nanoparticles on tissues. Since the main uptake of nanoparticles is by hepatic Kupffer cells and splenic macrophages, they have been described to induce hepatic dysfunction and structural changes related to, for example, increased levels of interleukin-1β and interleukin-6, pro-inflammatory cytokines, as well as increased alkaline phosphatase, liver enzyme aspartate aminotransferase (AST) and alanine aminotransferase (ALT) [123,124,125,126]. In addition, pyroptosis, lipid peroxidation (induced by reactive oxygen species), autophagy, and necrosis are pathways of hepatocyte damage and death attributed to nanoparticle toxicity in various studies [122,124,126].

However, specific differences make LnNPs a valuable and safe option to use in theranostics for cancer patients. First, the immune system does not recognize LnNPs as foreign elements because of the biocompatibility of their surface due to the use of biomolecules (Table 2). Second, chemical interactions that could form reactive oxygen species are minimized due to the steric effect of the biomolecules covering the nanoparticle. As previously demonstrated [6], the administration of functionalized lanthanide nanoparticles in mice does not show signs of inflammation (lymphocytic infiltration), hemorrhage, granuloma formation, fibrotic processes, or necrosis in healthy tissues [6]. Spleen and liver macrophages have been reported as the only mechanisms of cellular uptake of functionalized lanthanide nanoparticles [5]. Notably, macrophages are highly resistant to the point that they are able to repair double-strand breaks in DNA [124,127]. Increased transcriptional activation of NF-κB and Bcl-xL expression have been described as pro-survival mechanisms of macrophages [128]. In addition, macrophages damaged by irradiation or oxidative stress continue being metabolically active, viable, exhibit a reduced anti-inflammatory profile (decreased expression of mannose receptor C-type 1, versican, interleukin 10, and the cluster of differentiation 163, CD163) and increased phagocytic activity [127]. Phagocytosis of certain metal nanoparticles by macrophages has been reported to induce pro-inflammatory mechanisms [128]. However, in the case of functionalized lanthanides (e.g., Lu_2_O_3_-iPSMA/iFAP), no damage is observed in healthy tissues, which is attributed to biocompatibility mediated by ligands coating the nanoparticle surface [6].

As an interesting case, preclinical results have shown that Lu_2_O_3_-iPSMA/-iFAP nanoparticles activated by neutron irradiation inhibit colorectal HCT116 tumor progression due to a combination of radiotherapy, prolonged tumor retention, and molecular recognition of iPSMA and iFAP, which made possible the in vivo tumor optical and nuclear imaging. There was no liver and renal toxicity evidence since negligible uptake values in non-target tissues were observed. The first clinical case of a patient with colorectal liver metastases was the proof-of-concept showing the selective uptake of the LnNPs by liver metastases but not in the liver parenchyma (Figure 3) [5].

As mentioned above, luminescence imaging and drug delivery using LnNPs is another area where preclinical studies have demonstrated their theranostic potential. For example, the UCNPs@polyHPMA-5FU nanosystem is used for luminescence imaging and selective cancer drug delivery [37]. Also relevant is the development of target-specific biomolecules coated on Yb/Er/Ho enriched NaGdF_4_ nanoparticles as NIR-II systems for luminescence imaging, MR imaging, X-ray CT imaging, and photodynamic therapy (Table 2).

## 5. Conclusions and Outlook

Lanthanide-based nanoparticles for multimodal imaging and multifunctional therapy in cancer have been in discussion as potential nanomedicine tools for about two decades. However, to achieve their application in cancer patients, molecular targeting is mandatory. Functionalization must ensure that LnNPs selectively target the tumor microenvironment by directing them to one or more molecular targets. Once accumulated around or inside cancer cells, lanthanide nanosystems must be retained to effectively exert the therapy for which they were designed (photothermal therapy, radiotherapy, photodynamic therapy, chemotherapeutic drug release, etc.), as well as to detect, in time and space, molecular processes by different imaging modalities (optical imaging, nuclear imaging, magnetic resonance imaging, etc.).

Likewise, LnNPs should have a particle size between 10 and 100 nm, so that most of them can be captured and retained by tumors without crossing the endothelium of healthy tissues, while those NPs captured in a smaller percentage by macrophages (endothelial reticulum system) can be eliminated over time through the lymphatic system, avoiding damage to normal organs.

Analyzing the advances presented in this review, we can conclude and establish the following outlook:Many lanthanide-based nanosystems have successfully completed the preclinical phases, so their preparation under good manufacturing practices as stable colloidal dosage forms should be carried out to accelerate their clinical translation.Considering recent advances in cancer molecular biology, a greater number of different LnNPs, designed as multimodal and multifunctional nanosystems, should be functionalized with molecular targeting biomolecules associated with immunosuppression checkpoints and with those associated with tumor microenvironment remodeling. For example, functionalization with peptide inhibitors of PD-L1, PD-1, and FAP proteins would allow monitoring of disease prognosis and follow-up, as well as determining whether anti-PD-L1 immunotherapy can be combined with radiotherapy using neutron-activated lanthanide nanoparticles. It would also allow a better understanding of disease progression, therapeutic resistance, and metastasis in patients in a personalized manner.There is a need to increase the number of preclinical studies of lanthanide-based nanosystems whose imaging and therapeutic features could be evaluated with the imaging equipment and technology most widely available in different medical centers. This would allow LnNPs to become clinically translated in less time.Nevertheless, recent advances in the synthesis of monodisperse, biocompatible, and surface-functionalized LnNPs have been of paramount importance in demonstrating their usefulness in cancer theranostics, from which clinical applications could become a reality in the short and medium term.

## Figures and Tables

**Figure 1 nanomaterials-14-00296-f001:**
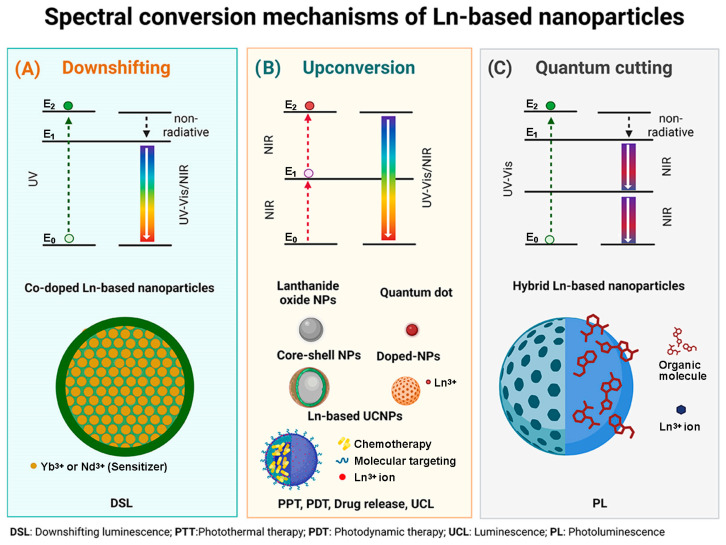
(**A**) Downshifting mechanism associated with Co-doped Ln-based NPs; (**B**) Upconversion mechanism associated with Ln_2_O_3_ NPs, QDs, Core–shell Ln-based NPs, Ln-doped NPs and multimodal Ln-based platforms, and (**C**) Quantum cutting mechanism associated with hybrid Ln-based nanoparticles.

**Figure 2 nanomaterials-14-00296-f002:**
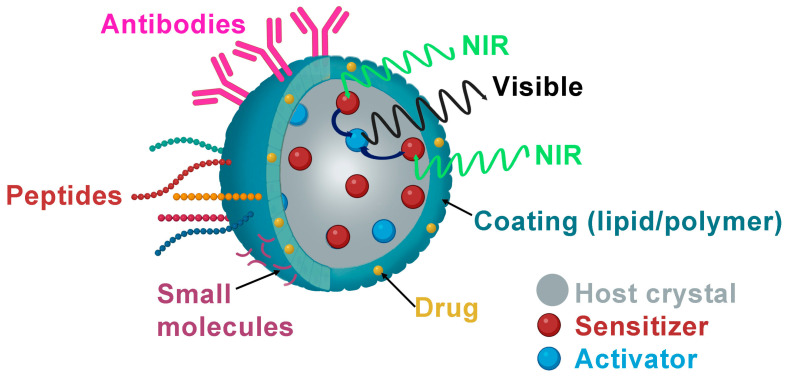
Schematic illustration of biological ligand-mediated targeted UCNPs that could be loaded with anticancer drugs as DOX and functionalized with small molecules as FA, peptides or antibodies.

**Figure 3 nanomaterials-14-00296-f003:**
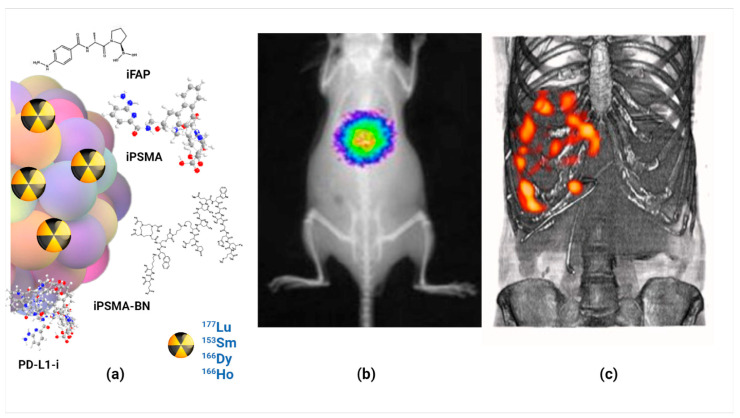
Schematic illustration of (**a**) molecularly targeted lanthanide nanoparticles for (**b**) optical imaging, (**c**) nuclear imaging and targeted radiotherapy (theranostics) of malignant tumors; observe the selective uptake of radiolanthanide lutetium oxide nanoparticles in colorectal liver metastases with negligible uptake in the liver parenchyma.

**Table 1 nanomaterials-14-00296-t001:** Multimodal imaging and therapy platforms based on upconversion nanoparticles (UCNPs)*^.^*

Design of Nanoparticles	Characteristics	Synthesis Method	General Applications	Examples of Systems with Applications in Cancer Disease	Ref.
UCNPs@SiO_2_ (a) Silica Shell (SiO_2_) (b) Mesoporous silica Shell (mSiO_2_)	− Good biocompatibility − Surface easily functionalization: COOH, NH_2_, SH, etc.	Stöber method for hydrophilic ligands.	PDT PTT Stimulus-response therapy. UCL	UCNPs@SiO_2_-NH_2_@FA/PEG/TCPP for PDT in vitro. PC-UCNPs@SiO_2_/siRNA/HA for siRNA delivery for therapy in vitro and in vivo. Gd-doped-NaYF_4_:Yb^3+^, Er^3+^@SiO_2_-NH_2_-Ab_2_ for UCL in vitro. UCNPs@SiO_2_-N-p-Tosylglycine/Ce6/GSH for PDT and stimulus-response therapy in vitro. UCNPs@mSiO_2_-Ce6-GPC3 for PDT in vitro and in vivo.	[18,35,36,37,38,39]
		Reverse micelle method for hydrophobic ligands.			
UCNPs@polymers	− Good water stability − Biocompatibility − Easy functionalization. − Adequate blood circulation times.	Wet chemistry methods	Multimodal image (UCL, MT, MRI). Drug nanocarrier	UCNPs@polyHPMA-5FU as drug nanocarrier.	[40]
UCNPs@noble metal nanocomposite (a) Au (b) Ag	Good water dispersibility.	Wet chemistry methods (a) Hydrothermal method (b) Thermal decomposition methods	UCL PTT	Ag-NaF_4_:Yb^3+^/Er^3+^@SiO_2_ nanocomposites for UCL in vivo. NaYF_4_:Yb^3+^/Er^3+^@SiO_2_ nanocomposites for UCL in vitro.	[41,42]

UCNP: upconversion nanoparticles; PDT: photodynamic therapy; PTT: photothermal therapy; UCL: upconversion luminiscence; MT: magnetization transfer; MRI: magnetic resonance imaging; FA: folic acid; TCPP: tetrakis(4-carboxyphenyl)porphyrin; siRNA: small interference RNA; HA: Hyaluronic acid; PC: polycations; Ce6: chlorine e6; GPC3: glypican 3 protein; GSH: glutathione; 5-FU: 5-fluorouracil; HPMA: N-(2-hydroxypropil)methacrylamide.

**Table 2 nanomaterials-14-00296-t002:** Targeted lanthanide-based nanoparticles for molecular imaging and therapy ^1^.

Type of Target Molecule	Type of Ligand	Nanoagent	Target Carcinoma	Applications	Ref.
Small Mol.	Folic Acid	AA-modified UCNPs	Human nasopharyngeal epidermal carcinoma (KB)	UCL in vitro	[73]
		UCNPs	Mouse melanoma (B16-F0)	NTr and PDT in vivo	[74]
		UCNC-Tm-FA, UCNC-Er-FA UCNC-Er, Tm-FA	Cervical cancer (HeLa)	UCL in vitro	[75]
		UCNPs-Er, Yb-FA	Ovarian cancer (CAOV3 cells)	Detect receptors in tissues	[76]
		UCNPs-Tb, Tm, Fe, NaGdF_4_	Cervical cancer (HeLa)	Multimodal imaging UCL, MRI, and Xray in vivo	[77]
		UCNPs-Nd NaGdF_4_	Cervical cancer (HeLa); Mouse liver cancer (H22)	PS and PDT in vivo	[78]
		LTA-UCNPs-Er, Yb	Mouse melanoma (B16-FO)	SDT, PDT, ROS and PTT in vivo	[79]
		UCNPs-PEG (DOX)	Folate receptors overexpressed in cancer cells	UCL and chemotherapy in vitro	[80]
		Caged UCNPs	Cervical cancer (HeLa)	UCL and chemotherapy in vivo	[81]
		UCNPs@MIL-53/FA	Cervical cancer (HeLa)	UCL and chemotherapy in vitro	[82]
		UCNPs-MoS(2)QDs	Cervical cancer (HeLa) Human liver cancer (HepG2)	UCL, PDT, chemotherapy in vitro	[83]
		MSNs DOX-NLPLNPs@MSNs	Human liver cancer (HepG2)	NIR luminescence and chemotherapy in vivo	[84]
		Ga:HfO_2_ Eu:HfO_2_ Tb:HfO_2_	Colon carcinoma (Caco-2)	luminescence, MRI and CT properties in vitro	[85]
		GdOF:Yb,Er-DOX&RB&FA	Cervical cancer (HeLa) Murine cervical carcinoma (U14)	UCL, MRI and CT images. PDT and chemotherapy in vivo	[54]
Peptides	Ang2	TLDoxCe6-NPs	Human glioblastoma (U87MG)	Metronomic PDT and chemotherapy in vivo	[86]
		Er-DCNPs-Dye-BP-ANG	Human glioblastoma (U87MG)	NIR IIb fluorescence imaging	[64]
		ANG-IMNPs	Murine brain tumor model, (ALTS1C1)	PDT and PTT in vivo	[87]
	Bombesin	UCNPs	Prostate tumors	UCL imaging, MRI, and CT	[88]
	iPSMA	Lu_2_O_3_ Dy_2_O_3_ Ho_2_O_3_	Hepatocarcinoma, prostate tumors	Optical imaging Nuclear imaging Radiotherapy	[23,53,89]
	KE108	UCNP-RB/PNBMA-PEG-AB3	human medullary thyroid cancer	UCL imaging and chemotherapy and PDT in vivo	[90]
	RGD	UCNPs Yb, Er,Tm	Human glioblastoma (U87MG)	UCL imaging in vivo	[91]
		UCNP-Gd-RGD	Human glioblastoma (U87MG)	UCL imaging and MRI	[68]
		rUCNPs@HSA(Ce6-Mn)	Human glioblastoma (U87MG)	MRI and PDT of glioma	[92]
		UCNP@TTD-cRGD NPs	Triple-negative breast cancer cells (MDA-MB-231)	ROS and PDT in vivo	[93]
		UNCP-ICG-TOS-RGDs	Human glioblastoma (U87MG)	UCL and CT imaging. PTT and chemotherapy in vivo	[94]
Peptides dual-targeting	Bombesin/iPSMA	Sm_2_O_3_	Prostate tumors, hepatocarcinoma	Optical imaging Nuclear imaging Radiotherapy	[23]
	iFAP/iPSMA	Lu_2_O_3_	Colorectal liver metastases	Optical imaging Nuclear imaging Radiotherapy	[5,6]
	RGD NGR	UCNP@P-RGD-NGR	Human lung cancer (A549)	UCL imaging in vivo	[95]
	Peptides targeting EBNA1 and LMP1	UCNP-P_n_, *n* = 5, 6, and 7	Cervical cancer (HeLa); Nasopharyngeal carcinoma (C666)	pH-sensitive imaging and immunotherapy in vivo	[96]
Antibodies	Anti-EGFR	Eu^3+^-doped CaF_2_ nanoparticles	Oral epithelial carcinoma (KB) Human epidermoid carcinoma (A431)	In vitro bioimaging Fluorescent imaging	[97]
	Anti-EGFR	Anti-EGFR-UNPs	Mouse ear tissue (Express EGFR)	Scanning microscopy	[98]
	AFP-specific antibodies	PEI-PLNPs (based on Eu^3+^ and Dy^3+^)	Cancer Cell Growth	Fluorescence Resonance Energy Transfer to detect AFP during cancer cell growth.	[99]
	mAb-201b	^177^Lu_0.5_Gd_0.5_(PO_4_)@Au@PEG800@Ab	Pulmonary metastatic disease	In vivo Radiotherapy MRI and SPECT imaging	[71]
	anti-CD33	antiCD33-LONp-PMI	Acute myeloid leukemia	In vitro fluorescent	[100]
	anti-PD-L1	ErNP	Murine colon cancer (CT-26 tumor)	In vivo immunotherapy Downconversion luminescence	[101]
RNA DNA Aptamers	siPD-L1	siPD-L1- NaGdF_4_	Breast and colon cancer	In vivo cancer treatment	[102]
	AS1411 aptamer	UCNPs@PDL@dsDNA/DOX	Lung cancer	Doxorubicin nuclear delivery	[103]

^1^ Upconversionluminescence (UCL); Nanotransducers (NTr); photosensitizers (PS); photodynamic therapy (PDT); reactive oxygen species (ROS); photothermal therapy (PTT); sonodynamic therapy (SDT); Persistent-Luminiscent nanoparticles (PLNPs).

**Table 3 nanomaterials-14-00296-t003:** Values of the zeta potential for the stability of nanocolloids.

Zeta Potential (mV)	Type of Force	Coloidal Stability
More than +/−61	Strong repulsive forces	Excellent
+/−40 to +/−60	Equilibrated repulsive force	Good
+/−30 to +/−40	Repulsive forces begin	Moderate
+/−10 to +/−30	Weak attractive forces	Unstable

## Data Availability

Not applicable.

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
