# Peer review of "Molecularly Targeted Lanthanide Nanoparticles for Cancer Theranostic Applications"

_nanomaterials, 2024, doi:10.3390/nano14030296_

Round 1

Reviewer 1 Report

Comments and Suggestions for Authors

G. Ferro-Flores, A. Ancira-Cortez, B. Ocampo-García and L. Meléndez-Alafort: Molecularly Targeted Lanthanide Nanoparticles for Cancer Theranostic Applications

The manuscript reports about the hot fields of biomedical application of lanthanides-containing nanosystems. The description of nanosystems is clearly articulated and with giving the concrete references, therefore the matter is a useful review. There are, however, some parts of the manuscript which are not written comprehensively or contain not right sentences, bad terms, and typos.

My comments, suggestions are as follows.

In Introduction

The abbreviation of tumor microenviroment (TME) must not change to TEM.

In 2. Physical properties of the lanthanide-based nanoparticles the features are concentrated to the optical characteristics. It should be mentioned that high difficulties of characterization appear in the smallest fractions of the range between 10 and 100 nm diameter. The Zeta potential characteristics are described in a later point. The 2.2. point must be corrected. PDT plays two times (lines, 168, 176), instead of PPT. Upconversion described first time, then Downshifting, while in Fig. 1 on the contrary. (that reminds me to Fig. of [42]). I suggest inserting the representation of states E0, E1, E2 also in Fig. 1. Ref [48] belongs to the next point. X-rays do not ionize typically; those are used for characterisation by scattering or absorption. I suggest changing the first sentence of point 2.4. How can the X-rays absorb incorrectly? I suggest distinguishing the characterisations used for revelation of physical, physical chemical, even in case of biological, biological distribution features and medical routine studies. The former ones are achieved by the means of state of art type equipment (research centres, synchrotron), while the latter are executed with simple tools.

In line. “They efficiently absorb excitation energy, which is then transferred to the lanthanide ion when it returns to its basal state. Consistent citation style and passive tone are employed throughout. (??, This sentence sounds a little bit strange.)

I suggest modifying and partly rewritten the paragraph 2.

In 3. In line 334, “Table 2” must be written. Although, Fig. 2 is a schematic illustration, the coating (lipid/polymer) shell of host crystal must be signed.

In the paragraph 4 the colloidal stability of LnNPs is mentioned and the values of zeta potential are discussed briefly. Indeed, the precursor stability is important, but the targeted LnNPs do not exhibit such a high absolute zeta potential value. The values are drastically reduced in presence of biological relevant buffer system or interstitial liquids.

Comments on the Quality of English Language

The quality of English language is correct.

Author Response

We thank the reviewer for their helpful and constructive criticism, which has improved the quality of our work. Our responses to the reviewer's comments are summarized below. Corrections have been highlighted in yellow in the manuscript.

The manuscript reports about the hot fields of biomedical application of lanthanides-containing nanosystems. The description of nanosystems is clearly articulated and with giving the concrete references, therefore the matter is a useful review. There are, however, some parts of the manuscript which are not written comprehensively or contain not right sentences, bad terms, and typos.

My comments, suggestions are as follows.

In Introduction

The abbreviation of tumor microenviroment (TME) must not change to TEM.

The correction was performed.

In 2. Physical properties of the lanthanide-based nanoparticles the features are concentrated to the optical characteristics. It should be mentioned that high difficulties of characterization appear in the smallest fractions of the range between 10 and 100 nm diameter. The Zeta potential characteristics are described in a later point.

The physicochemical characterization of lanthanide oxide nanoparticles included in this review can be achieved through techniques such as TEM, FT-IR, DLS, XRD, and Z potential. These techniques demonstrate that the characterization process is feasible and not more challenging than that of other nanoparticle systems.

The 2.2. point must be corrected. PDT plays two times (lines, 168, 176), instead of PPT.

The sentence was corrected

 Upconversion described first time, then Downshifting, while in Fig. 1 on the contrary. (that reminds me to Fig. of [42]). I suggest inserting the representation of states E0, E1, E2 also in Fig. 1.

The order of the text has been changed to match the order of the figures. Additionally, the reviewer's suggestion to include the representation of states E0, E1, and E2 has been implemented. Figure 1 is original.

Ref [48] belongs to the next point.

The reference was placed in the correct position

 X-rays do not ionize typically; those are used for characterisation by scattering or absorption. I suggest changing the first sentence of point 2.4. How can the X-rays absorb incorrectly?

The text was corrected as follows:

X-rays are a high-energy electromagnetic radiation with short wavelengths and high penetrating power.

 How can the X-rays absorb incorrectly?

The word correctly was eliminated from the next paragraph:

Materials with a high atomic number have demonstrated the greatest efficacy due to their ability to absorb X-rays

I suggest distinguishing the characterisations used for revelation of physical, physical chemical, even in case of biological, biological distribution features and medical routine studies. The former ones are achieved by the means of state of art type equipment (research centres, synchrotron), while the latter are executed with simple tools.

The physical characterization of lanthanide nanoparticles is what has been most studied throughout the modifications in their structure and functionalization, as described in the first section of this review.  However, the biological characterizations are, at this point, the most critical to achieve their clinical translation. This is what we have tried to highlight in the conclusions and perspectives of this review. The authors believe that distinguishing between the differences in physical and biological characterization is unnecessary and may be confusing without enriching the content and purpose of the manuscript.

In line. “They efficiently absorb excitation energy, which is then transferred to the lanthanide ion when it returns to its basal state. “Consistent citation style and passive tone are employed throughout”. (??, This sentence sounds a little bit strange). I suggest modifying and partly rewritten the paragraph 2.

The sentence has been corrected and the second paragraph has been rewritten as follows:

They efficiently absorb excitation energy, which is then transferred to the lanthanide ion when it returns to its basal state. The lanthanide ion uses this transferred energy to generate luminescence [44, 45].

If the sensitizing molecules are organic, their absorption efficiency is due to the presence of p bonds, and the energy is absorbed through pp* transitions. Conversely, if they are inorganic molecules (e.g., phosphonates), their absorption efficiency is mainly due to dipole-dipole, dipole-magnetic, or dipole-electric transitions [44, 46].

In 3. In line 334, “Table 2” must be written.

The correction has been made.

 Although, Fig. 2 is a schematic illustration, the coating (lipid/polymer) shell of host crystal must be signed.

The figure has been modified in agreement with the reviewer’s suggestion.

 In the paragraph 4 the colloidal stability of LnNPs is mentioned and the values of zeta potential are discussed briefly. Indeed, the precursor stability is important, but the targeted LnNPs do not exhibit such a high absolute zeta potential value. The values are drastically reduced in presence of biological relevant buffer system or interstitial liquids.

In agreement with the reviewer and to clarify this point, the following paragraph was added:

The stability of nanoparticles as a pharmaceutical form with high zeta potential values is a prerequisite for their in vivo application. However, interactions with components of biological fluids, such as proteins, electrolytes, and small molecules, influence their electrical potential and reduce the absolute value of the zeta potential, although without affecting the stability of the nanosystems when coated with biomolecules covalently bound to their surface, which guarantees colloidal stability [23].

Reviewer 2 Report

Comments and Suggestions for Authors

I carefully read this review and found it suitable to be published in the Nanomaterial journal. The review refers to the use of lantadine nanoparticles in theranostics and especially for cancer, based on their special optical, magnetic, or radiation properties, and their small size in the 10-100 nm range. The manuscript is well structured and fits on the specifics of the journal, and contains enough relevant references that are quite recent.

However, I would recommend the authors to complete their manuscript with more relevant figures to support the results presented in the literature.

Also, for tables 1 and 2 and figures 1 and 2, their sources (references) must be specified and at the same time, the copyrights must be indicated, if they are not original (of the authors).

Author Response

I carefully read this review and found it suitable to be published in the Nanomaterial journal. The review refers to the use of lantadine nanoparticles in theranostics and especially for cancer, based on their special optical, magnetic, or radiation properties, and their small size in the 10-100 nm range. The manuscript is well structured and fits on the specifics of the journal, and contains enough relevant references that are quite recent.

However, I would recommend the authors to complete their manuscript with more relevant figures to support the results presented in the literature.

We consider the three figures to be representative of the results presented for the content of the article. For example, Figure 1: physicochemical properties, Figure 2: functionalization with biomolecules and Figure 3 represents the theranostic applications and the translational process emphasized in this review. The authors believe that the inclusion of specific references would not represent the overall content and results of this research.

Also, for tables 1 and 2 and figures 1 and 2, their sources (references) must be specified and at the same time, the copyrights must be indicated, if they are not original (of the authors).

All tables and figures included in this review are originals.

Reviewer 3 Report

Comments and Suggestions for Authors

The authors present a very extensive, detailed, and clear review regarding Nanoparticles containing Lanthanide species for theranostics applications.

The review covers several aspects and approaches which exploit these nanoparticles, as well as the most relevant physical mechanisms present in this family of nanoparticles.

I recommend this manuscript for publication after some minor revisions.

1)      Line 30, doped should be replaced by dopant. It is not the ion that is doped but the nanoparticles.

2)      Lines 47 and 52,  the acronym TEM is a typo. I think the authors meant TME.

3)      In Line 98, the use of theranostic is sloppy. Theranostic, as written in this line is an adjective, hence the sentence is confusing. The authors should write something like:

“A theranostic molecule possesses unique and versatile properties that make it 98 suitable for use in both therapy and diagnosis.”

4)      Line 107, the ionizing radiation emitted will depend on the isotope formed, hence the authors should make it clearer. As it is written it seems that all the resulting isotopes will have beta particle radiotherapy.

5)      Line 248, the first sentence should be improved to something as:

“X-rays are a type of ionizing electromagnetic radiation with short wavelengths and high penetrating power.”

This because X-rays are not waves of EM radiation, but the EM radiation waves themselves. Furthermore, It is obvious that the penetration power refers to the matter, since EM radiation have no attenuation in empty space. It is obvious that the authors know that, but the phasing was a bit slopy.

6)      Line 295, the expression

“(..)the emitting and receiving electrons must have the same spin (process spin allowed) to produce fluorescence emission.”

Should be replaced by something like

“(…) the emitting and receiving electrons must conserve their spin moment to enable fluorescence emission.”

7)      Lines 422 and 536, the word “theragnostic” should be replaced by “theranostic”. Even though the expression “theragnostic” and “theragnostics” can be found in the literature, for a question of consistency of this manuscript, only one of the terms should be employed. In this case the most used one. theranostic.

8)      Line 426, what do the authors mean by “such as least leaching reduction“? This is confusing and should be clarified.

9)      All over the text there is a constant problem with the superscripts and subscripts that must be corrected, namely (but not limited) in the electronic configuration fillings and the chemical formulas, respectively.

10)    Finally, even though I consider that the optical properties are quite well covered, the authors briefly mentioned the applicability of the magnetic properties of the lanthanide-based NPs. It would be very fruitful to add a section (even if small) regarding this aspect of these NPs.

Author Response

We appreciate Reviewer constructive criticism, which has enhanced the quality of our work. Please find our responses to the reviewer's comments below. Corrections have been highlighted in blue in the manuscript.

 The authors present a very extensive, detailed, and clear review regarding Nanoparticles containing Lanthanide species for theranostics applications. The review covers several aspects and approaches which exploit these nanoparticles, as well as the most relevant physical mechanisms present in this family of nanoparticles.

 I recommend this manuscript for publication after some minor revisions.

1)      Line 30, doped should be replaced by dopant. It is not the ion that is doped but the nanoparticles

The word has been corrected.

 2)      Lines 47 and 52, the acronym TEM is a typo. I think the authors meant TME.

Text has been corrected.

 3)      In Line 98, the use of theranostic is sloppy. Theranostic, as written in this line is an adjective, hence the sentence is confusing. The authors should write something like:

“A theranostic molecule possesses unique and versatile properties that make it 98 suitable for use in both therapy and diagnosis.”

The sentence has been rewritten in accordance with the reviewer’s suggestion.

 4)      Line 107, the ionizing radiation emitted will depend on the isotope formed, hence the authors should make it clearer. As it is written it seems that all the resulting isotopes will have beta particle radiotherapy.

The text has been rewritten as follows:

Neutron activation produces radioluminescence imaging with possible radiotherapy applications when beta particles are emitted.

 5)      Line 248, the first sentence should be improved to something as:

“X-rays are a type of ionizing electromagnetic radiation with short wavelengths and high penetrating power.”

This because X-rays are not waves of EM radiation, but the EM radiation waves themselves. Furthermore, It is obvious that the penetration power refers to the matter, since EM radiation have no attenuation in empty space. It is obvious that the authors know that, but the phasing was a bit slopy.

The sentence has been rewritten in accordance with the reviewer’s suggestion.

 6)      Line 295, the expression

“(..)the emitting and receiving electrons must have the same spin (process spin allowed) to produce fluorescence emission.”

Should be replaced by something like

“(…) the emitting and receiving electrons must conserve their spin moment to enable fluorescence emission.”

The sentence has been rewritten as suggested by the reviewer.

 7)      Lines 422 and 536, the word “theragnostic” should be replaced by “theranostic”. Even though the expression “theragnostic” and “theragnostics” can be found in the literature, for a question of consistency of this manuscript, only one of the terms should be employed. In this case the most used one. theranostic.

The word has been replaced.

 8)      Line 426, what do the authors mean by “such as least leaching reduction“? This is confusing and should be clarified.

The sentence has been rewritten as follows:

The Gd:HfO2 NPs showed the best properties, such as the lowest degradation rate and no relevant in vitro cytotoxicity.

 9)      All over the text there is a constant problem with the superscripts and subscripts that must be corrected, namely (but not limited) in the electronic configuration fillings and the chemical formulas, respectively.

The nomenclature has been reviewed throughout the manuscript and has been corrected accordingly.

 10)    Finally, even though I consider that the optical properties are quite well covered, the authors briefly mentioned the applicability of the magnetic properties of the lanthanide-based NPs. It would be very fruitful to add a section (even if small) regarding this aspect of these NPs.

The following paragraph has been added to cover magnetic properties of NPs.

2.6 Magnetic Resonance Imaging

MRI is based on the net polarization of the nuclear spins of water protons in the presence of an intense magnetic field of 1.5 to 3T. Therefore, the magnetic properties of the contrast agent require the insertion of one or more metal centers with unpaired electrons [66]. Gadolinium, terbium, dysprosium, and holmium oxide nanoparticles are of particular interest because they have appreciable magnetic moments, which is useful for MRI [67]. Gd3+ ions have been widely used to produce molecular T1 contrast enhancement [66]. Many papers have been published on Gd3+ based nanostructures of different composition, shape, and size. However, only a few of them have been functionalized with small molecules, such as folic acid [50,68], RGD, chlorotoxin and transactivator of transcription (TAT) peptides [69-71], and the anti-thrombomodulin antibody [72]. Other Ln3+ ions increase the magnetic moment as the magnetic field strength increases before reaching saturation, resulting in higher transverse relaxivities [73].

Tb2O3, Ho2O3, and Dy2O3 nanoparticles have been reported useful as T2 MRI con-trast agents at high MR fields in vivo [73,74]. Although the lanthanide ion Eu2+ also has seven unpaired electrons in its outer electron shell, it has a larger ionic radius than Gd3+, which results in a faster water exchange rate. This characteristic ensures that Eu-based contrast agents have relatively high relaxivity values [66]. However, to date, no functionalized nanoparticles containing Tb+3, Ho+3, Dy+3, and Eu2+ ions have been described for molecularly targeted MRI.

Round 2

Reviewer 1 Report

Comments and Suggestions for Authors

The authors corrected the manuscript and considered all comments and suggestions. There is a single remark; some letters are very small in Figure 1. I suggest enlarging them.

Author Response

Figure 1 has been corrected as suggested by the reviewer.